# A Metabolomic Approach and Traditional Physical Assessments to Compare U22 Soccer Players According to Their Competitive Level

**DOI:** 10.3390/biology11081103

**Published:** 2022-07-25

**Authors:** João Pedro da Cruz, Fábio Neves dos Santos, Felipe Marroni Rasteiro, Anita Brum Marostegan, Fúlvia Barros Manchado-Gobatto, Claudio Alexandre Gobatto

**Affiliations:** 1Laboratory of Applied Sport Physiology—LAFAE, School of Applied Sciences, University of Campinas, UNICAMP, Limeira, São Paulo 13484-350, Brazil; jpdacruz97@hotmail.com (J.P.d.C.); felipemarroni@hotmail.com (F.M.R.); anita_brum@hotmail.com (A.B.M.); fgobatto@unicamp.br (F.B.M.-G.); 2Institute of Chemistry, University of Campinas, UNICAMP, Campinas, São Paulo 13083-970, Brazil; fabiof6@gmail.com

**Keywords:** metabolomics, metabolic profile, anthropometric, critical velocity, hematocrit, soccer

## Abstract

**Simple Summary:**

Different physical tests are applied in the sports context for athlete selection and training prescriptions. Recently, the metabolomics approach has been used to understand biological systems; it has shown high potential to answer integrated questions, including with regard to soccer. Our study is the first to combine traditional physical assessment protocols with metabolomics to detail and compare the physiological profiles of two soccer teams (under-22 soccer players) at different competitive levels. Using anthropometric measurements and the critical velocity model, no differences were observed between elite and non-elite soccer players. However, the metabolomic analysis was able to differentiate among the fasting serum metabolic profiles of athletes of different competitive levels. Our results suggest that the application of metabolomics combined with traditional physical assessment protocols can improve characterizations of athletes, strengthening the differentiation of the physiological profiles of soccer teams. Thus, the metabolomic approach can contribute by offering additional data in individual characterizations of athletes, improving training programs, and consequently, increasing the performance of players during competitions.

**Abstract:**

The purpose of this study was to use traditional physical assessments combined with a metabolomic approach to compare the anthropometric, physical fitness level, and serum fasting metabolic profile among U22 soccer players at different competitive levels. In the experimental design, two teams of male U22 soccer were evaluated (non-elite = 20 athletes, competing in a regional division; elite = 16 athletes, competing in the first division of the national U22 youth league). Earlobe blood samples were collected, and metabolites were extracted after overnight fasting (12 h). Untargeted metabolomics through Liquid Chromatograph Mass Spectrometry (LC-MS) analysis and anthropometric evaluation were performed. Critical velocity was applied to determine aerobic (CV) and anaerobic (ARC) capacity. Height (non-elite = 174.4 ± 7.0 cm; elite = 176.5 ± 7.0 cm), body mass index (non-elite = 22.1 ± 2.4 kg/m^2^; elite = 21.9 ± 2.3 kg/m^2^), body mass (non-elite = 67.1 ± 8.8 kg; elite = 68.5 ± 10.1 kg), lean body mass (non-elite = 59.3 ± 7.1 kg; elite = 61.1 ± 7.9 kg), body fat (non-elite = 7.8 ± 2.4 kg; elite = 7.3 ± 2.4 kg), body fat percentage (non-elite = 11.4 ± 2.4%; elite = 10.5 ± 1.7%), hematocrit (non-elite = 50.2 ± 4.0%; elite = 51.0 ± 4.0%), CV (non-elite = 3.1 ± 0.4 m/s; elite = 3.0 ± 0.2 m/s), and ARC (non-elite = 129.6 ± 55.7 m; elite = 161.5 ± 61.0 m) showed no significant differences between the elite and non-elite teams, while the multivariate Partial Least Squares Discriminant Analysis (PLS-DA) model revealed a separation between the elite and non-elite athletes. Nineteen metabolites with importance for projection (VIP) >1.0 were annotated as belonging to the glycerolipid, sterol lipid, fatty acyl, flavonoid, and glycerophospholipid classes. Metabolites with a high relative abundance in the elite group were related in the literature to a better level of aerobic power, greater efficiency in the recovery process, and improvement of mood, immunity, decision making, and accuracy, in addition to acting in mitochondrial preservation and electron transport chain maintenance. In conclusion, although classical physical assessments were not able to distinguish the teams at different competitive levels, the metabolomics approach successfully indicated differences between the fasting metabolic profiles of elite and non-elite teams.

## 1. Introduction

Physical capacity, morphological characteristics, and tactical and technical skills differentiate soccer players by competitive level [1]. Increased aerobic and anaerobic capacities, VO_2max_, muscle strength and power, running speed, agility, and lower percentage of body fat have been identified as essential characteristics in elite players compared to other competitive levels (i.e., sub-elite, amateur, recreational) [1]. Considering these important aspects, distinct traditional physical assessments such as anthropometric and body composition measurements [2,3], lactate minimum test [4], maximal lactate steady state [5], maximum accumulated oxygen deficit, [6] and critical velocity (CV) [7,8] have been applied to soccer players for talent evaluation, competitive level discrimination, and load training prescription and control [1].

Besides studies involving protocols with traditional physical assessments, traditional biochemistry has been used to determine specific biological markers in soccer players [9]. Hematological parameters and markers such as complete blood count, aspartate aminotransferase, alanine aminotransferase, creatine kinase, creatinine, urea, uric acid, cholesterol, ferritin, C-reactive protein, and thyroid-stimulating hormone have become part of routine laboratory tests applied to elite soccer athletes [10]. Although traditional biochemistry has enabled the differentiation of high-level athletes from other competitive levels, this traditional analysis is limited when the objective is to seek a global and integrated response of metabolites. In this regard, untargeted metabolomics can be an interesting strategy to investigate the physiologic profile of athletes, since this approach aims to provide a broad and integrated perspective of an organism’s metabolic pathways [11]. Metabolomic analysis can be conducted by liquid chromatograph mass spectrometry (LC-MS), nuclear magnetic resonance (NMR), gas chromatography mass spectrometry (GC-MS), and capillary electrophoresis mass spectrometry (CE-MS), or in combination with other analyses [12] in order to determine the metabolic profile of athletes using biological fluids (urine, saliva, sweat, and blood) [13].

This approach involves the study of metabolite profiles in comparative situations of groups submitted to different interventions, such as therapies, stress levels, dietary modulation, diseases and physical exercise [11,14]. Despite the difficulty in sample preparation, instrumental availability, complexity of data analysis, and reduced practical dissemination of the metabolomics approach in soccer, Al-Khelaifi, et al. [15] studied the profile of dietary metabolites present in the serum of elite athletes of various sports, including soccer. In addition, changes in the metabolic profile were investigated after soccer matches [16,17,18,19,20], acute and chronic physical training [21,22,23,24], and application of evaluation protocols [25]. However, untargeted metabolomic studies have not yet been used to differentiate the metabolic profile of soccer players at different competitive levels.

In this context, the combination of traditional physical assessments and metabolomics to characterize and differentiate soccer athletes may greatly contribute to the study of exercise and sports physiology. Therefore, this study aimed to use typical physical assessments together with a metabolomics approach to compare the anthropometric profile, physical fitness level, and serum fasting metabolic profile among U22 soccer players from different competition levels. It was hypothesized that U22 soccer players at higher levels of competition would present better parameters for anthropometric profile, physical fitness level, and different metabolic profile than non-elite U22 soccer players.

## 2. Methods

### 2.1. Participants

After a detailed explanation about the risks and objectives of the study, written informed consent was signed by all participants. Thirty-six soccer players from two U22 teams playing at different levels of competition were evaluated, that is, a non-elite group (n = 20 athletes, age = 20 ± 2 years) competing in the São Paulo regional division and an elite group (n = 16 athletes, age = 18 ± 1 years) competing in the first division of the São Paulo Soccer Championship. Both teams were in the second week of preseason. During the trial period, the teams had access to meals provided by the clubs (without control of the amount of individual food). The participants were instructed to keep the same hydration habits and avoid hard physical activity as well as alcohol and caffeine ingestion. Subjects who reported using ergogenic resources, smoking and/or having osteoarticular problems were excluded. The Physical Activity Readiness Questionnaire (PAR-Q) [26] was adopted for safety. The protocol was approved by the Research Ethics Committee of The School of Medical Sciences—University of Campinas-UNICAMP (CAAE—15540619.6.0000.5404).

### 2.2. Design

Six evaluation sessions were conducted in the present study (Figure 1). Initially, the athletes answered the anamnesis and the Physical Activity Readiness Questionnaire (PAR-Q), and were instructed to perform a 12-h overnight fast. In the second session, blood samples from the subjects’ earlobe were collected for untargeted metabolomic analysis and hematocrit determination. In addition, the individuals underwent anthropometric assessments. In the third, fourth, fifth, and sixth sessions, CV was applied to determine aerobic and anaerobic capacities.

### 2.3. Blood Collection

After a 12-h overnight fast, four capillary blood samples from the individuals’ earlobe were collected and centrifuged at 11,000× *g* for 5 min (Microprocessor Centrifuge for Tubes—Q222TM) at room temperature. Soon after serum separation, the samples were stored in 1.5 mL cryogenic tubes and subjected to immediate freezing in liquid nitrogen. Afterward, the serum samples were cryopreserved in a freezer at −80 °C for LC-MS analysis. To determine the hematocrit, one additional capillary sample was collected, centrifuged, and analyzed using a graduated ruler [27].

### 2.4. Anthropometric Assessments

For the measurements of all variables, the procedures of the International Society for the Advancement of Kinanthropometry [28] were adopted. The somatotype was determined by the method proposed by Heath and Carter [29] from anthropometric variables, which allowed the estimation of three morphological components: endomorph, ectomorph, and mesomorph. All skinfolds were measured in triplicate by the same evaluator, using the average of the values obtained from the three measurements. To determine the percentage of body fat, four skinfolds (triceps, suprailium, subscapula, and abdomen) measured with the aid of a Lange skinfold caliper were used [30]. The equation below was used to determine the percentage of fat, where Σ4SKF refers to the sum of the skinfold thicknesses:% Body fat = Σ^4^SKF × 0.153 + 5.783

### 2.5. Physical Fitness Level Assessment

The athletes performed four predictive trials (shuttle runs—20 m) at total distances of 800, 1200, 1600, and 2000 m, randomly, with an interval of 24 h between each effort on a soccer field. The times (tlim) obtained for each of the requested distances were recorded using a stopwatch. To determine CV and anaerobic running capacity (ARC) (aerobic and anaerobic variables, respectively), an individual graph was plotted containing the values of distance (meters) on the *y*-axis and time (seconds) on the *x*-axis. The mathematical model adopted to adjust the data was linear, that is, distance versus tlim [7]. CV and ARC, respectively, were obtained from the slope and intercept on the *y*-axis of the regression line. The R^2^ data from the mathematical model were considered the main result of the application feasibility analysis of the CV protocol.

### 2.6. Serum Samples Preparation for Metabolomics

The extraction of serum lipids was carried out using a biphasic solvent system of cold methanol, methyl tert-butyl ether (MTBE), and water with some modifications [31]. Specifically, 225 μL of cold methanol was added to a 50 μL blood serum aliquot in a 1.5 mL Eppendorf tube, followed by vortexing (20 s). Subsequently, 750 uL of cold MTBE was added, also followed by vortexing (20 s). Afterward, all aliquots were vortexed for 1 min. Phase separation was induced by adding 200 μL of Milli-Q water. After vortexing (20 s), the samples were centrifuged at 14,000× *g* for 10 min at 4 °C (Eppendorf Centrifuge 5427 R). The upper organic phase was evaporated through manual drying using nitrogen gas (N60 AIR LIQUID, 99.9999%). Dried extracts were resuspended using 100 μL of a mixture of acetonitrile and isopropanol (ACN/IPA—60:40, *v*/*v*) followed by vortexing (30 s).

### 2.7. LC-MS Data Acquisition and Metabolite Identification

The LC-MS analyses were based on the parameters used by Cajka and Fiehn [32]. Serum lipids were analyzed using an UltiMate™ 3000 Standard Binary System LC system (Thermo Scientific™, Waltham, MA, USA) coupled to an Orbitrap Q Exactive™ Focus mass spectrometer (Thermo Scientific™, Waltham, MA, USA) equipped with an electrospray source (ESI). The 2 μL serum lipid extract was injected into the LC-MS equipment. The temperature of the sampler was maintained at 6 °C. Serum lipids separated in Kinetex column (1.7μ F5 100 A 100 × 2.1 mm) were analyzed in ESI (+), while the same lipid extract separated in Acquity UPLC^®^ Beh (C8 1.7 μ 2.1 × 100 mm) was analyzed in ESI (−). Both columns were used at 65 °C, with a flow rate of 0.6 mL/min. The mobile phases consisted of (A) 60:40 (*v*/*v*) acetonitrile: water with 10 mM ammonium formate; and (B) 90:10 (*v*/*v*) isopropanol: acetonitrile with 10 mM ammonium formate. The separation was conducted under the following gradients: 0 min: 5% (B); 0–15 min: 95% (B); 15–22 min: 95% (B); 22–22.1 min: 5% (B); and 22.1–27 min: 5% (B). For both ESI polarities the parameters adopted were: desolvation gas flow, 55; auxiliary gas flow, 15; sweep gas flow, 3; spray voltage, 3.50 kV; capillary temperature, 275 °C; RF Lens S, 50; and auxiliary gas temperature, 450 °C. The mass spectrometer acquisition mode was set to Full-MS. The mass range was set at 100–1500 Da, with MS resolution at 70,000, AGC target of 3e6 and maximum IT of 200 ms. Data acquisition was performed using Thermo Scientific TraceFinder software. The data were processed on the online version of XCMS software [33] to correct retention times, peak alignment, and quantitative feature extraction. The major features were annotated respecting level 2 [34] by searching exact masses in the Metlin, HMDB, Keeg, LipidMaps, and MINE spectra databases using CEU Mass Mediator 3.0 software [35]. All annotated metabolites were within 10 ppm of their exact masses. 

### 2.8. Univariate and Multivariate Statistical Analysis

Mean and standard deviation (SD) were calculated for all the studied variables. The normality and homogeneity of anthropometric data, hematocrit percentage, CV and ARC were confirmed by the Shapiro-Wilk and Levene tests, respectively. These data were compared using the unpaired *t*-test (*p* < 0.05). Effect size (ES) was calculated according to Cohen’s categories: small if 0 ≤ |d| ≤ 0.5; medium if 0.5 < |d| ≤ 0.8; and large if |d| > 0.8) [36]. Confidence intervals were also calculated for standard deviation with α = 0.05 (σ/√n). Metaboanalyst 5.0 was used to perform the metabolomic statistical treatment [37]. Data were filtered by the relative standard deviation, normalized by the sum and scaled using pareto scaling. The first stage of the analysis consisted of applying the fold change (FC) analysis (fold change threshold: 2) to identify features with different relative abundances between the soccer teams. Then, the metabolites that showed differences were submitted to principal component analysis (PCA), aiming to provide a quick visualization of similarities or differences in the metabolite dataset. Subsequently, partial least squares discriminant analysis (PLS-DA) was used, allowing the distinction between clusters of elite and non-elite teams based on metabolomic data. To validate this analysis, cross-validation for optimal number of components for classification was used. The maximum components were found to be five. The classification was made based on the 10-fold method. Accuracy indices, Q^2^ (model predictability), and r^2^ (explained variance) were used as a performance measure of PLS-DA. Discrimination between the physiological profiles of both teams was performed using a variance importance in projection (VIP) score greater than 1.0, estimated from the PLS-DA analysis.

## 3. Results

### 3.1. Anthropometric, Body Composition, Aerobic and Anaerobic Characteristics

The general anthropometric and body composition characteristics of all subjects separated by teams, including height, body mass index, body mass, lean body mass, body fat, and hematocrit, are shown in Table 1. There were no significant differences between the non-elite and elite groups, and the variables exhibited small effect sizes (Table 1). In addition, the non-elite group showed a prevalence of 80% of mesomorph and 20% of ectomorph, while in the elite group these percentages were 87% and 13%, respectively. No endomorphs were found in our sample.

The data from the application of the critical velocity protocol are presented in Table 2. Regarding the times to complete the predictive trials, CV, ARC, and r^2^, there were no differences between the non-elite and elite teams, and the variables exhibited small effect sizes.

### 3.2. Metabolites with Differences between the Soccer Teams

After processing the raw data in XCMS, 10,230 features were extracted, of which 5022 were extracted in ESI mode (+) and 5,208 in ESI mode (−).

The processing data in Metaboanalyst indicated major features from the LC-MS analysis in both ionization modes. First, features from the non-elite and elite groups were compared (Figure 2). The FC analysis revealed that 76 metabolites showed different relative abundances between the teams for the ESI (+) mode. For the non-elite players, there was a prevalence of 49 features, while the elite team presented 27 features. On the other hand, 64 features were different for the ESI (−) mode, of which 39 showed greater relative abundances for the non-elite group and 25 for the elite group (Appendix A).

Figure 3 shows the unsupervised multivariate analysis (PCA). In this score plot, the *x*- and *y*-axes were named PC 1 and 2, respectively, and each point represents 1 soccer player. Although the PCA score plots reveal low cumulative explained variance, they provide a clear separation tendency for both the ESI (+) (Figure 3A) and ESI (−) modes (Figure 3B), which was further confirmed in the PLS-DA analysis, besides the absence of outlier samples. 

Supervised multivariate analysis of PLS-DA was applied to reveal the clusters of the elite and non-elite teams (Figure 4). This mathematical model showed accuracy = 0.98, R^2^ = 0.88 and Q^2^ = 0.81 for the ESI (+) mode and accuracy = 0.99, R^2^ = 0.86 and Q^2^ = 0.79 for the ESI (−) mode, indicating the validity of the cross-validation of the model. In the PLS-DA score plot for the ESI (+) (Figure 4A) and ESI (−) modes (Figure 4C), a clear separation of the samples can be observed in the two clusters. The VIP score plot shows the analysis of the 15 most essential features from the whole data, with a score above 1 for both ESI (+) (Figure 4B) and ESI (−) (Figure 4D). 

### 3.3. Annotation of Metabolites

Feature annotation is the assignment of potential metabolite candidates to the signal based on the correspondence between their mass and database or library entries. The identified features were annotated and named putative metabolites. The most essential baseline metabolites from the VIP score plot are shown in Table 3. Nineteen metabolites were annotated from the 30 discriminatory features between the teams (Appendix A). The relative abundance of the metabolites is shown in the VIP score plot (Figure 4B,D). The blue and red color spectrum boxes on the right indicate the relative abundance values of the corresponding possible metabolites in each team. Of the metabolites annotated, all of them showed greater relative abundance for the elite team, with the exception of PS (20:1(11Z)/16:0), which was relatively more abundant in the non-elite team.

## 4. Discussion

Contrary to our hypotheses, anthropometric variables, body composition, hematocrit, predictive trials, and aerobic and anaerobic capacities were not significantly different between the elite and non-elite teams. However, the metabolomic analysis was capable to distinguish the fasting metabolic profile of the evaluated teams. Furthermore, this methodology made it possible to identify the main metabolite classes responsible for differentiating the non-elite and elite U22 soccer teams. Although researchers have been characterizing metabolic changes after the application of physical tests at the end of matches and during physical preparation in soccer pre- and full seasons [16,17,22,23], our study was the first to combine traditional physical assessment protocols and metabolomics to detail and compare the physiological profile of two soccer teams at different competitive levels.

Historically, traditional physical assessments provide parameters that contribute to the knowledge of the essential characteristics required for an athlete to play soccer matches at a highly competitive level [38]. Anthropometric assessments are applied to determine physical condition and adaptation to training, in addition to discriminating the competitive level of athletes. In soccer, the Heath and Carter [29] method has been used to determine somatotypes (mesomorph, endomorph and ectomorph). Studies have demonstrated the prevalence of mesomorphism in elite, non-elite, adult, and youth soccer players [39,40,41]. Corroborating the literature, our results showed that there was a prevalence of 80% and 87% of mesomorph in the non-elite and elite teams, respectively. Additionally, body composition plays a key role in the physical constitution of professional soccer players [1]. In the present study, the general characteristics of body composition, including height, body mass index, body mass, lean body mass, and body fat, did not show statistical differences between the teams. Likewise, Ostojic [42] found that height, weight, and the sum of seven skinfolds were not different between the elite and non-elite groups. In contrast, Arnason, et al. [43] observed that elite league teams presented statistically greater differences than lower-league teams. Furthermore, body fat percentage was shown to be higher in sub-elite than in elite soccer players [3].

The hematocrit refers to the percentage of red cells in the total blood volume, being important to identify and diagnose anemia, malnutrition, dehydration, or excessive hydration [27]. In soccer players, the hematocrit showed significant changes at the end of the season compared to the beginning [10]. This tendency was maintained after 6 weeks of soccer training [44]. In this study, the teams did not present statistical difference in the percentage of hematocrit, corroborating the study conducted by Ostojic [45], which did not indicate differences in the percentage of hematocrit among players at different competitive levels.

During a 90-min match, soccer players run approximately 10 km at an average intensity close to the anaerobic threshold, a parameter related to aerobic capacity [38]. There are several traditional physical assessments responsible for determining the aerobic capacity of soccer players [4,5]. CV was capable to detect adaptations in soccer players of both genders at different competitive levels after training periods [8,46,47]. However, our aerobic and anaerobic capacity data from the CV test did not discriminate between non-elite and elite players. In general, no difference was observed when comparing the variables derived from the traditional physical assessments of both teams. These findings could be explained by season moment, evaluated category, and soccer competitive level of the country. It is worth mentioning that our athletes were in the second week of the preseason, while in other studies they were in the middle and end of the season [3,43]. Here, the category evaluated was U22, unlike some studies in the literature (which investigated, for example, U16 and professional teams) [3,43]. Furthermore, Brazil has a highly competitive level, regardless of the division. It is important to point out that in São Paulo, the training programs in this phase (preseasons) for the category studied appear to be similar to those for teams at different competitive levels, which may explain the non-difference in anthropometric and performance parameters evaluated. Despite that, the metabolomic analysis was able to distinguish the physiological profile of the evaluated teams at rest. 

In order to direct the biological inference, FC, PCA, and PLS-DA analyses were used to identify which metabolites presented different relative abundances between the teams. In other studies, these mathematical models showed excellent performance in detecting metabolic changes after different situations and interventions in soccer [20,22,23]. In this study, both PCA and PLS-DA analyses enabled the construction of score plots revealing a clear separation between soccer players, evidencing that the metabolic profile of non-elite and elite teams were significantly different. In particular, the PLS-DA analysis helped in the annotation of some lipid metabolites such as saringosterol 3-glucoside (glycerolipids), 3b,5a,6b-Cholestanetriol (sterol lipids), neuromedin N(1-4) (fatty acyls), PG(P-20:0/16:0) (glycerophosphoglycerols) and FAHFA(18:3-(2-O-24:0)). Furthermore, this mathematical model demonstrated that other metabolites belonging to the flavonoid and glycerophospholipid classes are more abundant in the serum of elite athletes compared to non-elite athletes.

Flavonoids (cycloartomunin) are a class of metabolites found in fruit, thus being commonly consumed in the human diet [48]. Nieman et al. [49] showed that individuals with increased concentrations of flavonoids are more effective in combating exercise-induced inflammation and oxidative stress. On the other hand, glycerophospholipids are fundamental elements for cell membranes that contribute to a variety of metabolic processes and intracellular signaling [50]. The most important ones are phosphatidylserine (PS), phosphatidylcholine (PC), phosphatidylethanolamine (PE), and cardiolipin (CL). PS accounts for 10% of all membrane glycerophospholipids. In one study, physically active males showed improved exercise capacity due to a greater concentration of PS in the body [51]. In addition, Starks et al. [52] demonstrated that increased PSs are effective in combating exercise-induced stress and preventing physiological deterioration. In general, PSs act on human recovery and performance, also improving mood, immunity, decision-making and accuracy [53]. Other metabolites such as PCs and PEs are the most abundant glycerophospholipids in the cell membrane. The literature shows that individuals with high maximal aerobic power (VO_2_max) had higher serum concentrations of PCs and PEs compared to individuals with low VO_2_max [54,55], making these metabolites possible biomarkers of VO_2_max level. In addition, PEs play an important role in mitochondrial function and morphology, also contributing to the autophagy process [56,57,58]. Like PEs, CLs are present in cells with a large number of mitochondria, being essential for the mitochondrial energy metabolism and the maintenance of the electron transport chain [59]. 

Strengths and limitations must be taken into account. Considering that this investigation is an observational (cross-sectional) study based on a specific cohort of participants (soccer players of the U22), causal relationships and extrapolation of our findings to other populations must be carefully performed, as suggested by Castro, et al. [60]. On the other hand, we emphasize that our findings were characterized by standardized physiological exams applied in a specific environment for the modality. For example, the CV protocol [7] was highly successful in determining the aerobic capacity of the athletes (R^2^ = 0.99 ± 0.01). Another interesting aspect to consider is the blood collection from the earlobe of individuals, making it more accessible and painless for them [61]. Furthermore, our exploratory metabolomic analysis was based on a standardized and robust technique (LC-MS) for the identification of a large number of metabolites [62], increasing the possibility of signaling new possible metabolites to identify differences between elite and non-elite soccer players. In future opportunities, it would be important to investigate the responsiveness of the metabolites appointed by our study in front of physical training for elite and non-elite groups [63].

Finally, elite and non-elite teams that apparently presented equality in physical conditioning, in reality showed relevant differences from a metabolomic point of view. Metabolomics expanded the understanding of the metabolic profile of these players. The elite team presented more metabolites than the non-elite team, some of them being possible biomarkers of increased VO_2_max [54,55]. In this study, some PCs and PEs possibly indicate greater aerobic power of elite players, since the VO_2_max was considered one of the most powerful discriminators among male soccer players at different competitive levels [1]. In addition, other discriminating metabolites can bring some benefits, such as potentiation of the recovery process, increased mood, immunity, decision making and accuracy, and improved exercise capacity with mitochondrial preservation and electron transport chain maintenance [52,53,56,57,58,59]. Thus, the metabolomic approach can contribute by offering additional data on the individual characterization of athletes, improving training programs, and consequently increasing the performance of players during competitions. 

## 5. Conclusions

In the present study, traditional physical assessments were not able to differentiate soccer players at different competitive levels. However, the metabolomic approach successfully distinguished the fasting metabolic profile of elite and non-elite teams, showing that the elite soccer players presented an increased number of serum lipid metabolites belonging to the glycerolipid, sterol lipid, fatty acyl, flavonoid, and glycerophospholipid classes. Therefore, this study suggests that the application of metabolomics combined with classic physical assessment protocols can improve the characterization of athletes, strengthening the differentiation of the physiological profile of soccer teams. Additionally, this approach may contribute to the emergence of new research lines for soccer scientists aiming to unravel the complexity of this modality from different perspectives, such as the identification of new biomarkers for monitoring fatigue throughout the season, physical adaptation to training, and maturation of new talents. Lastly, it could provide promising information on the performance of elite soccer players.

## Figures and Tables

**Figure 1 biology-11-01103-f001:**
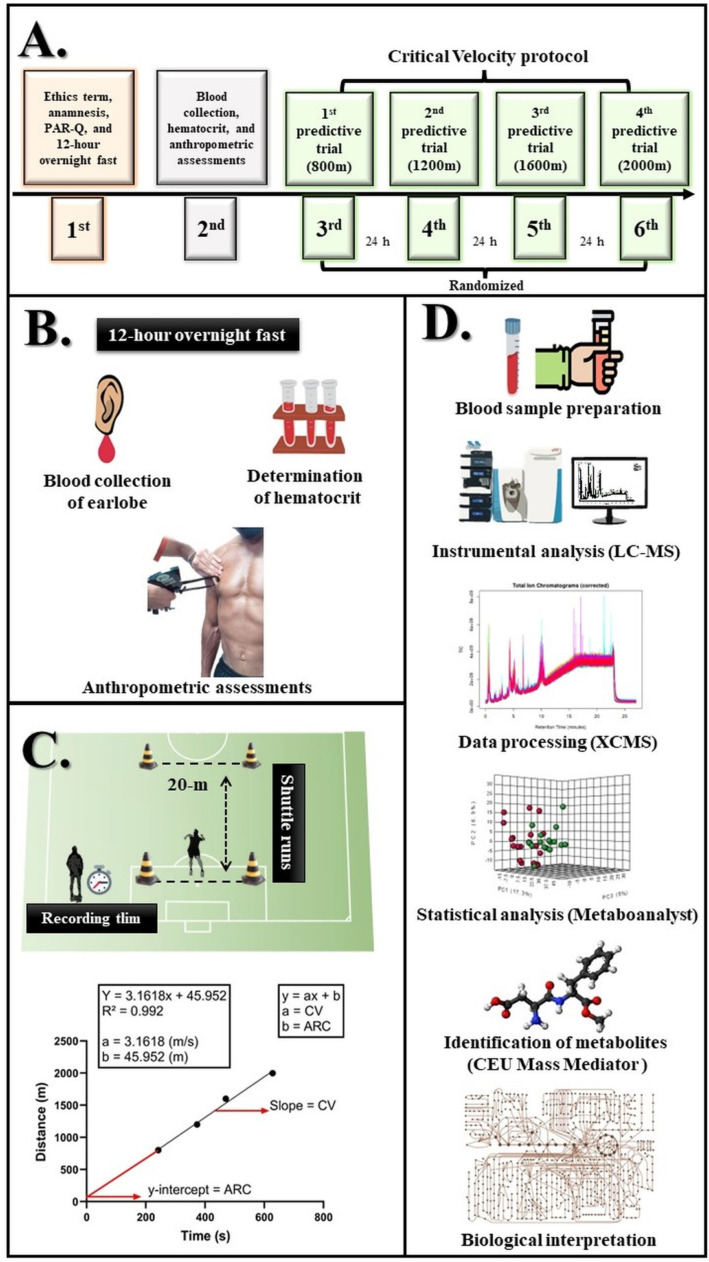
(**A**) shows the experimental design used in the study. (**B**) exemplifies blood sample collection after an overnight fast for metabolomic analysis and volume determination of red blood cells (i.e., hematocrit). Besides blood collection, anthropometric assessments were carried out (body composition by skinfold measurement and somatotype analysis). (**C**) shows the application of the critical velocity protocol in the field for the determination of aerobic (CV) and anaerobic (ARC) capacities. (**D**) illustrates the workflow for the liquid chromatography mass spectrometry (LC-MS)-based metabolic profile analysis of serum specimens from the soccer players.

**Figure 2 biology-11-01103-f002:**
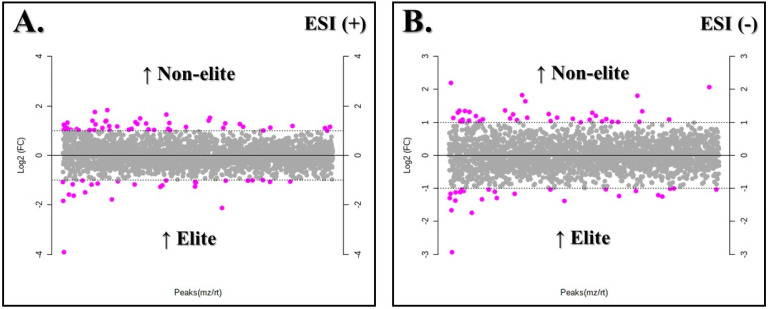
Comparison of the physiological profiles of soccer players from both elite and non-elite teams for the ESI (+) (**A**) and ESI (−) modes (**B**) by fold-change analysis with threshold 2. The pink circles represent different features between the teams. The upper pink circles represent features above the threshold with greater relative abundance for the non-elite team, while the lower pink circles represent features above the threshold with greater relative abundance for the elite team.

**Figure 3 biology-11-01103-f003:**
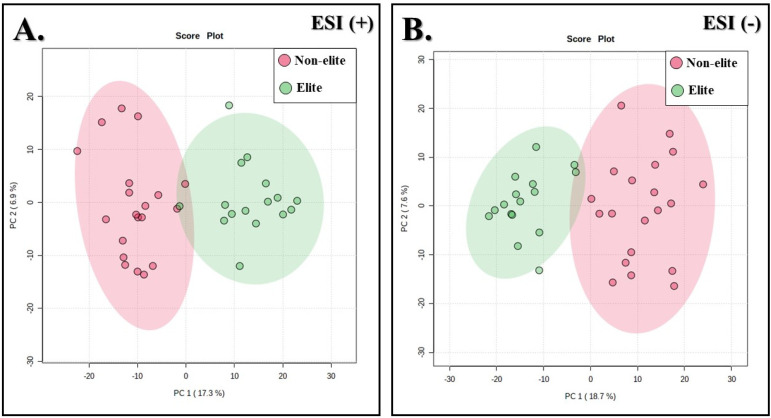
PCA score plots of all serum samples from both soccer teams for the ESI (+) (**A**) and ESI (−) modes (**B**). Colors represent different soccer teams (red: non-elite athletes; green: elite athletes).

**Figure 4 biology-11-01103-f004:**
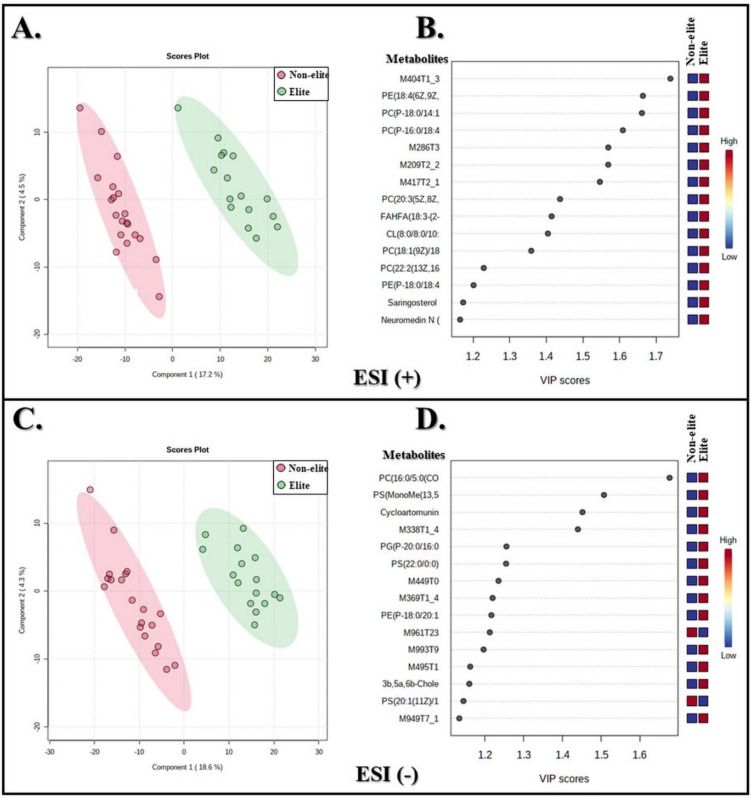
(**A**,**C**): 2D PLS-DA score plots comparing both groups in the ESI (+) and ESI (−) modes, respectively. (**B**,**D**): PLS-DA score plots showing the main features that discriminate the sample groups according to VIP scores for the ESI (+) and ESI (−) modes, respectively. The colors represent different soccer teams (red: non-elite athletes; green: elite athletes). The blue and red color spectrum boxes on the right indicate the values of relative abundances of the corresponding features in each team. Features M404T1_3, M286T3, M209T2_2, M417T2_1 in ESI (+) mode and M338T1_4, M449T0, M369T1_4, M961T23, M993T9, M495T1, M949T7_1 in ESI (−) mode were not identified.

**Table 1 biology-11-01103-t001:** Characterization of anthropometric and body composition variables of both teams.

			Statistical Analysis	
	**Non-Elite** **(n = 20)** **Mean ± SD** **(CI)**	**Elite** **(n = 16)** **Mean ± SD** **(CI)**	**Comparison**	**ES**
**Height** **(cm)**	174.4 ± 7.0 (5.1–10.8)	176.5 ± 7.0 (5.3–10.2)	t = −0.95 *p* = 0.348	0.30
**BMI ** **(kg/m^2^)**	22.1 ± 2.4 (1.7–3.7)	21.9 ± 2.3 (1.7–3.3)	t = 0.16 *p* = 0.870	0.09
**Body mass (kg)**	67.1 ± 8.8 (6.5–13.6)	68.5 ± 10.1 (7.6–14.7)	t = −0.45 *p* = 0.654	0.15
**Lean body mass** **(kg)**	59.3 ± 7.1 (5.2–10.9)	61.1 ± 7.9 (6.0–11.5)	t = −0.74 *p* = 0.462	0.24
**Body fat** **(kg)**	7.8 ± 2.4 (1.7–3.7)	7.3 ± 2.4 (1.8–3.5)	t = 0.53 *p* = 0.597	0.21
**BF** **(%)**	11.4 ± 2.4 (1.7–3.7)	10.5 ± 1.7 (1.2–2.4)	t = 1.28 *p* = 0.207	0.44
**Hematocrit (%)**	50.2 ± 4.0 (2.9–6.1)	51.0 ± 4.0 (3.0–5.8)	t = −0.58 *p* = 0.563	0.20

**BMI**—body mass index; **BF**—body fat percentage; **SD**—standard deviation; Values in parentheses represent the upper and lower confidence limits (CI) of standard deviations; **Comparison** (*p* < 0.05)—*t*-test for independent samples; **ES**—effect size.

**Table 2 biology-11-01103-t002:** Predictive trials and aerobic and anaerobic capacities of both teams from the application of the critical velocity protocol.

			Statistical Analysis	
	**Non-Elite** **(n = 20)** **Mean ± SD** **(CI)**	**Elite** **(n = 16)** **Mean ± SD** **(CI)**	**Comparison**	**ES**
**1st predictive trial (s)**	232.0 ± 15.0 (11.1–23.2)	232.0 ± 16.0 (12.1–23.3)	t = 0.08 *p* = 0.933	0.01
**2nd predictive trial (s)**	371.0 ± 26.0 (19.2–40.2)	366.0 ± 26.0 (19.7–37.9)	t = 0.50 *p* = 0.616	0.19
**3rd predictive trial (s)**	495.0 ± 33.0 (24.3–51.1)	497.0 ± 20.0 (15.2–29.2)	t = −0.20 *p* = 0.841	0.08
**4th predictive trial (s)**	627.0 ± 56.0 (41.3–86.6)	656.0 ± 43.0 (32.7–62.8)	t = −1.69 *p* = 0.099	0.59
**CV ** **(m/s)**	3.1 ± 0.4 (0.3–0.6)	3.0 ± 0.2 (0.1–0.3)	t = 1.98 *p* = 0.06	0.33
**ARC** **(m)**	129.6 ± 55.7 (41.1–86.21)	161.5 ± 61.0 (46.3–89.1)	t = −1.63 *p* = 0.110	0.55
**R^2^**	0.98 ± 0.02 (0.01–0.03)	0.99 ± 0.01 (0.01–0.02)	t = −0.57 *p* = 0.569	0.67

**1st predictive trial**—total distance performed: 800 m; **2nd predictive trial**—total distance performed: 1200 m; **3rd predictive trial**—total distance performed: 1600 m; **4th predictive trial**—total distance performed: 2000 m; **CV**—aerobic capacity (critical velocity); **ARC**—anaerobic capacity (anaerobic running capacity); **R^2^**—Regression coefficient from the linear adjustment, i.e., distance versus time; **SD**—standard deviation; Values in parentheses represent the upper and lower confidence limits (CI) of standard deviations; **Comparison** (*p* < 0.05)—*t*-test for independent samples; **ES**—effect size.

**Table 3 biology-11-01103-t003:** Features annotated and selected by the analysis of VIP scores as the most capable of differentiating the elite and non-elite teams.

Feature	*m*/*z*	RT (min)	Putative Metabolite	Mode	Chemical Formula	Error ppm	VIP Score
			**Glycerophospholipids**				
			** *Phosphatidylserines* **				
M889T7	888.53	6.75	PS(MonoMe(13,5)/DiMe(9,5))	−	C_49_H_82_NO_12_P	5	1.50
M580T1_3	580.36	1.42	PS(22:0/0:0)	−	C_28_H_56_NO_9_P	2	1.25
M789T6_2	788.54	5.83	PS(20:1(11Z)/16:0)	−	C_42_H_80_NO_10_P	2	1.14
			** *Glycerophosphoglycerols* **				
M798T6	797.55	6.10	PG(P-20:0/16:0)	−	C_42_H_83_O_9_P	7	1.25
			** *Phosphatidylethanolamines* **				
M740T4	739.54	3.76	PE(18:4(6Z,9Z,12Z,15Z)/P-18:1(11Z))	+	C_41_H_72_NO_7_P	10	1.66
M803T7_1	802.59	6.98	PE(P-18:0/20:1(11Z))	−	C_43_H_84_NO_7_P	6	1.21
M747T4_1	746.51	3.62	PE(P-18:0/18:4(6Z,9Z,12Z,15Z))	+	C_41_H_74_NO_7_P	2	1.20
			** *Phosphatidylcholines* **				
M608T1_3	608.35	0.91	PC(16:0/5:0(COOH))	−	C_29_H_56_NO_10_P	3	1.67
M755T3	754.52	2.81	PC(P-18:0/14:1(9Z))	+	C_40_H_78_NO_7_P	10	1.66
M721T2	720.53	2.34	PC(P-16:0/18:4(6Z,9Z,12Z,15Z))	+	C_42_H_76_NO_7_P	1	1.61
M856T7_1	855.66	6.55	PC(20:3(5Z,8Z,11Z)/20:1(11Z))	+	C_48_H_88_NO_8_P	2	1.43
M787T9	786.60	9.04	PC(18:1(9Z)/18:1(9Z))	+	C_44_H_84_NO_8_P	9	1.35
M909T2	908.61	2.33	PC(22:2(13Z,16Z)/22:6(4Z,7Z,10Z,13Z,16Z,19Z))	+	C_52_H_88_NO_8_P	2	1.22
			** *Cardiolipins* **				
M1087T1_1	1086.65	0.63	CL(8:0/8:0/10:0/18:2(9Z,11Z))	+	C_53_H_98_O_17_P_2_	5	1.40
			**Glycerolipids**				
M591T1	591.42	0.86	Saringosterol 3-glucoside	+	C_35_H_58_O_7_	6	1.17
			**Flavonoids**				
M447T1_1	447.14	0.79	Cycloartomunin	−	C_26_H_24_O_7_	6	1.45
			**Sterol Lipids**				
M465T3_2	465.35	3.45	3b,5a,6b-Cholestanetriol	−	C_27_H_48_O_3_	8	1.15
			**Fatty Acyls**				
M527T1_2	527.28	0.52	Neuromedin N (1-4)	+	C_26_H_40_N_4_O_6_	6	1.16
			**Other metabolites**				
M628T7	627.56	6.86	FAHFA(18:3-(2-O-24:0))	+	C_42_H_76_O_4_	6	1.41

***m*/*z***—m stands for mass and z stands for charge number of features; **RT**—Retention time. The metabolite class is represented in bold, while the subclass is represented in bold and italics in the putative metabolite column.

## Data Availability

Not applicable.

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
