# Peer review of "A Metabolomic Approach and Traditional Physical Assessments to Compare U22 Soccer Players According to Their Competitive Level"

_biology, 2022, doi:10.3390/biology11081103_

Round 1

Reviewer 1 Report

In “A Metabolomics Approach And Classical Physical Assessments 2 To Compare U22 Soccer Players According To Their Competitive Level”, João Pedro da Cruz et al. investigated the relationship between the “elite” status in U22 soccer players and various metabolites and other measurements. The authors used an experimental design which is most reminiscent of a case-control observational study. In general, the authors seem to use epidemiologic terms very liberally. Further, small sample size is presented and no replication is attempted for the findings.

Major

As a case-control study, with data collected at one point, it is impossible to determine which occurred first, metabolite changes or “elite” status acquisition, which limits applications of current study.

Table 3 is currently cut-off, hence, can not be assessed.

Abstract, discussion: the authors mention “high sensitivity in distinguishing the fasting metabolic profile of the evaluated teams”, but no sensitivity measures are provided (if by sensitivity, the authors meant the probability that when the outcome “elite” is present, the metabolite test (for nineteen metabolites) is positive).

From the discussion, it is not clear how the study results should be used to strengthen the differentiation of the physiological profile of soccer teams – should each of the selected 19 metabolites assessed individually for each player? Should some threshold level-fold change for each metabolite be used? Should they be assessed in a set? I believe, this is important, especially given low number of selected metabolites.

Small sample size and lack of an attempted replication is a concern for this study. For example, J. Westerhuis et al. (https://link.springer.com/article/10.1007/s11306-007-0099-6?utm_source=getftr&utm_medium=getftr&utm_campaign=getftr_pilot) advocates against the use of PLS-DA score plots for inference of class differences.

Minor

Abstract: please write out the PLS-DA (partial least squares discriminant analysis).

Line 90-91 – a number seems to be missing (CAAE – 91 xxxxxxxx.x.xxxx.xxxxx).

Figure 4 - need to re-name the graphs, so it would be clear in the figure legend which figure is described. That is, A. B. (instead of “A.”) and C. and D. (instead of “B.”)

Figure 4 – it is not clear from the figure legend what “features” are listed on the y-axis of the graphs to the right. If it is metabolites. Please consider listing metabolite name, instead of the “feature” name. For example, for M608T1_3, use “PC(16:0/18:4(6Z,9Z,12Z,15Z))”, as listed in Table 3.

In general, please use “feature name” in supplementary table only, and use “Putative metabolite” as your primary metabolite name throughout the article, including in tables/figures.

Not clear - how many metabolites were measured, overall?

Discussion: line 262-264: presume that the authors meant “not statistically significantly different between the elite and non-elite participants”.

Reviewer 2 Report

General comments:

This is a very clean and well thought out study design with many complexities. The presentation of the rationale, methods, and findings is very clearly laid out. I commend the authors on a very interesting study and manuscript.

Introduction: 

No significant comments on the introduction; existing literature is clearly laid out and the application/rationale for an integrated metabolomics approach is well established. I caution adding uncertainty within the introduction, but there may be benefit to a brief statement recognizing the practical limitation in a broader setting so that this issue can be addressed within the discussion.

Figure 1: It would help to make Panel A larger but realize constraints of space. I don’t have a specific recommendation but encourage the authors to consider a potential revision.

Table 1 and Table 2: (Small but potentially very helpful) Both are tough to read/follow horizontally. I would need to check the journal guidelines but shading and/or thinner lines to further differentiate between rows (groupings of mean+/-SD and CI) would help. 

Figures 3 and 4 provide a lot of context but the cumulative percentage of variance in pc[1,2] is relatively low. I don’t necessarily think it’s essential that full output from pca is shown, but at a minimum, this should be addressed/discussed somewhere within the results or discussion. 

Colorbars on Figure 4 do not reference any gradients within the figure. Please double check whether the colorbar itself was an error or revise the plots if the gradient was meant to be applied to the dots within the figures on the right-hand side. 

Contrary to what has been observed in the past, the authors did not observe a difference in CV between elite and non-elite groups. To me, this is very interesting and makes the findings from the metabolomics data increasingly relevant and novel. However, as to help future readers, I recommend that the authors provide a little bit of context/discussion around the degree of “non-elite” within this study relative to other published works that have observed a difference in various anthropometric and performance metrics. 

I urge caution in response to this question because I do not believe that such screening should become a determinant of contract awards and/or negotiation, but what is the practical application of the metabolomics approach within sport and/or research? Potentially a brief discussion around the ability/need to investigate how these data can contribute to individualized approaches. Even the addition of a single sentence to the conclusion would suffice.

Reviewer 3 Report

The topic is interesting and provide some new information in the literature, but, since some results are in total disagreement with the literature, a better interpretation of the results is needed. Therefore, a revision is required.

Line 18. Please specify the term LC-MS

Line 23. Please specify the term PLS-DA

Line 82. Please specify the importance of the information “ mesomorph=80 %, ectomorph=20 %” in this context. The somatotype has not been mentioned before. What does this information add in this part of the manuscript? It seems to be a result and is more appropriate in that section; the same for the information in Lines 83-84 .

Line 92. Please add the information regarding the Research Ethics Committee of the University

Line 95. Please specify “PAR-Q “

Line 98. Please replace the term  “submitted “with subjects “underwent” anthropometric assessment

Anthropometric assessment: please describe the anthropometric measurements before the calculation of body composition.

Line 114. I immagine that the instrument used was Lange caliper. If it is correct, please replace the term adipometer with caliper.

Figure 1. Please replace “anthropometric evaluation was measured” with “anthropometric evaluation was carried out”

Line 197. Please delete the comma after aerobic.

The conclusions need to be improved. It is very inusual not to find any kind of differences in anthropometric parameters, body composition, aerobic and anaerobic characteristics. A better interpretation of the results is necessary.

A Rebelo 1J BritoJ MaiaM J Coelho-e-SilvaA J FigueiredoJ BangsboR M MalinaA Seabra. Anthropometric characteristics, physical fitness and technical performance of under-19 soccer players by competitive level and field position nt J Sports Med. 2013 Apr;34(4):312-7. doi: 10.1055/s-0032-1323729. Epub 2012 Oct 11.

Sergej Ostojic. Elite and Nonelite Soccer Players: Preseasonal Physical and Physiological Characteristics. August 2010, Research in Sports Medicine An International Journal April-June 2004(2):143-150

Reviewer 4 Report

This study uses classical physical assessments and a metabolomic approach to compare the anthropometric, physical fitness level, and serum fasting metabolic profile among U22 soccer players from different competitive levels. The data in the figures and tables is clearly described, and the experiment design is reasonable. I think this manuscript is ready to accept after some minor revisions.

1.      At line 282, the article said that the two groups of body compositions showed no significant difference, which is different from the results of other researchers cited in line285-287. Can you further explain the possible reasons for the discrepancy?

2. At line 342-345, it said the benefits that others differentiated metabolites can bring. Please provide evidence that they can bring these benefits.

Round 2

Reviewer 1 Report

The authors have addressed most of the reviewer's prior comments.

This reviewer still believes that it would be useful to put the mean levels/SDs for the 19 metabolites (considered most useful in discriminating between the groups) for both elite and non-elite participants into a separate (supplementary) table.

Figure 4 can still use some improvement for clarity.

Otherwise, the manuscript has been improved.

Reviewer 3 Report

The Authors have taken into account all the observations I have made; thus in this form the manuscript is acceptable for publication.
